# Regioselective Ni-Catalyzed reductive alkyl-silylation of acrylonitrile with unactivated alkyl bromides and chlorosilanes

Jinwei Sun[1,2], Yongze Zhou[1,2], Rui Gu[1,2], Xin Li[1], Ao Liu[1] & Xuan Zhang [1]✉

Transition-metal catalyzed carbosilylation of alkenes using carbon electrophiles and silylmetal (-B, -Zn) reagents as the nucleophiles offers a powerful strategy for synthesizing organosilicones, by incorporating carbon and silyl groups across on C-C double bonds in one step. However, to the best of our knowledge, the study of silylative alkenes difunctionalization based on carbon and silyl electrophiles remains underdeveloped. Herein, we present an example of silylative alkylation of activated olefins with unactivated alkyl bromides and chlorosilanes as electrophiles under nickel catalysis. The main feature of this protocol is employing more easily accessible substrates including primary, secondary and tertiary alkyl bromides, as well as various chlorosilanes without using pre-generated organometallics. A wide range of alkylsilanes with diverse structures can be efficiently assembled in a single step, highlighting the good functionality tolerance of this approach. Furthermore, successful functionalization of bioactive molecules and synthetic applications using this method demonstrate its practicability.

Organosilicon compounds have continuously received more and more attentions over the past decades because of their unique properties and broad applications in a variety of areas. For instance, as a bioisostere of carbon atom in medicine[1–5], silicon has different physical and electronic properties in a number of aspects, which result in low toxicity, increased lipophilicity, significantly tuned polarization and favorable metabolic profiles compared with its carbon analogue. On the other hand, silicon-containing molecules have impressive applications in advanced hybrid materials[6–8] and surface modification[9–11]. Furthermore, organosilanes are important building blocks for organocatalysts[12–14] and selective C-C bond formations[15–17]. As a result, developing C-Si bond formation reactions that incorporate silicon atoms into organic molecules is not only a prerequisite but also a driving force to unlock the full potential of organosilicon compounds.

In this context, many useful and efficient methods including nucleophilic substituted reaction[18–20], cross-coupling reaction[21–32], carbene insertion into silanes[33–37], C-H bond silylation[38–45] and difunctionalization of unsaturated carbon-carbon bonds[46–52] for the

construction of carbon-silicon bond have been developed. Among them, transition-metal catalyzed silylative functionalization of alkenes with silicon reagents has emerged as one of most powerful strategies for synthesis of silicon-containing molecules. Hydrosilylation of alkenes enabled by transition metal catalysis[53–57] is one of the most investigated areas with significant progresses during last decades. However, this mono-functionalization largely reduces the complexity of possible molecular scaffolds, and often suffers from regioselectivity issues. In contrast, transition metal catalyzed carbosilylation of olefins is more attractive, but still in its infancy[58–65]. In earlier approaches, carbosilylation of alkenes is normally carried out in the presence of Grignard reagent[63–65], which involves nucleophilic silylation of in-situ generated alkyl magnesium species with chlorosilanes. However, the reactive intermediate restricted the functional group compatibility. Until recently, arylsilylation, alkenylsilylation and silylacylation of C-C double bonds using Si-M (M = B, Zn) reagents were reported by Eagle[61], Yin[60], and Brown[59] groups, respectively (Fig. 1a). Although these methods show better functional group tolerance than earlier reports,

[1]School of Chemistry and Materials Science, Institute of Advanced Materials and Flexible Electronics (IAMFE), Nanjing University of Information Science and Technology, 219 Ningliu Road, Nanjing 210044, China. [2]These authors contributed equally: Jinwei Sun, Yongze Zhou, Rui Gu. ✉e-mail: xuanzhang@nuist.edu.cn

**Fig. 1 | Transition-metal catalyzed carbosilylation of alkenes. a** Arylsilylation, alkenylsilylation and silylacylation of alkenes using silyl nucleophiles; **b** alkylsilylation of alkenes using silyl nucleophiles; **c** This work: alkylsilylation of alkenes using silyl electrophiles. The silyl groups are colored in pink.

the air- and moisture-sensitive silyl-metal reagents are difficult to prepare and handle. To overcome above problems, a class of silyl reagents—solid and salt-stabilized silylzinc pivalates with good stability was developed by Li and co-workers[58] very recently, and which was demonstrated unique reactivity in the alkylsilylation of alkenes under nickel catalysis (Fig. 1b). Unfortunately, only primary alkyl iodides and activate benzyl bromides are amenable substrates in this system. Undoubtedly, developing a new strategy using more readily available substrates for the construction of organosilanes based on carbosilylation of alkenes with improved structural complexity and diversity would be in high demand.

Noteworthily, all aforementioned methods are using the combination of carbon electrophile and silyl nucleophile as the reaction partners. As an alternative strategy, transition-metal catalyzed reductive difunctionalization of C-C double bonds using two electrophiles has recently attracted more attentions and provided a facile tool for construction of C-C bond and C-heteroatom bond in one step[66–74]. These reactions generally take place with the assistance of stoichiometric reductant to avoid the preparation and handling of organometallic reagents required in normal coupling processes, thereby show more better substrate compatibility. The development of silylative transformation of alkenes using carbon and silyl electrophiles to forge C-C and C-Si bonds would not only enrich methods for reductive

difunctionalization of olefins, but also provide a good opportunity for organosilicon chemistry. Herein, we present a regioselective nickel-catalyzed alkylsilylation of acrylonitrile using unactivated alkyl bromides and chlorosilanes as electrophiles (Fig. 1c). Although this type of product can also be achieved through nucleophilic substituted reaction of deprotonated alkyl nitrile with silyl chloride, organolithium compound used as the base leads the reaction exhibiting poor functional group tolerance and restricted application. In this work, we report an approach for synthesizing organosilicon compounds, and present a straightforward way to access alkylsilanes bearing versatile structural complexity.

## Results and discussion
### Reaction development
To initiate our study, we chose (3-bromopropyl)benzene (**1a**) as the carbon electrophile, acrylonitrile (**2a**) as the alkene unit, and chlorodimethyl(vinyl)silane (**3a**) as the silyl electrophile to investigate the optimal conditions. A variety of nitrogen-containing ligands were firstly explored using NiBr$_2$•DME as the catalyst, activated zinc dust as the reductant in $N$, $N$-dimethylacetamide (DMA) at 25 °C for 24 h. To our delight, when 2-(pyridin-2-yl)−4,5-dihydrooxazole (**L4**) was selected as the ligand, the reaction proceeded regioselectively to give desired product 2-(dimethyl(vinyl)silyl)−6-phenylhexanenitrile (**4**) in

**Table 1 | Optimization of reaction conditions[a]**

| Entry | Ni Cat. | Ligand | Reductant | Solvent | Yield of 4 (%) | Recovery of 1a (%) |
|---|---|---|---|---|---|---|
| 1 | NiBr₂•DME | L₁ | Zn | DMA | 0 | 77 |
| 2 | NiBr₂•DME | L₂ | Zn | DMA | 0 | 87 |
| 3 | NiBr₂•DME | L₃ | Zn | DMA | 6 | 73 |
| 4 | NiBr₂•DME | L₄ | Zn | DMA | 30 | 52 |
| 5 | NiBr₂•DME | L₅ | Zn | DMA | 0 | 97 |
| 6 | NiBr₂•DME | L₆ | Zn | DMA | 14 | 79 |
| 7 | Ni(acac)₂ | L₄ | Zn | DMA | 15 | 83 |
| 8 | Ni(OTf)₂ | L₄ | Zn | DMA | 25 | 55 |
| 9 | Ni(COD)₂ | L₄ | Zn | DMA | 12 | 54 |
| 10 | Ni(PPh₃)₂Cl₂ | L₄ | Zn | DMA | 35 | 47 |
| 11 | Ni(PPh₃)₂Cl₂ | L₄ | Zn | DMSO | 0 | 70 |
| 12 | Ni(PPh₃)₂Cl₂ | L₄ | Zn | CH₃CN | 0 | 94 |
| 13 | Ni(PPh₃)₂Cl₂ | L₄ | Zn | EtOAc | 0 | 88 |
| 14[b] | Ni(PPh₃)₂Cl₂ | L₄ | Zn | DMA | 60 | 21 |
| 15[b,c] | Ni(PPh₃)₂Cl₂ | L₄ | Zn | DMA | 55 | 63 |
| 16[b] | Ni(PPh₃)₂Cl₂ | L₄ | Mn | DMA | 4 | 64 |
| 17[b] | Ni(PPh₃)₂Cl₂ | L₄ | TDAE | DMA | trace | 72 |
| 18[b,d] | Ni(PPh₃)₂Cl₂ | L₄ | Zn | DMA | 79 (73[e], 55[f]) | 0 |

*DME* dimethyl ether, *DMA N,N*-dimethylacetamide, *acac* acetylacetonate, *OTf* triflate, *COD* 1,5-cyclooctadiene, *DMSO* dimethyl sulfoxide, *EtOAc* ethyl acetate, *TDAE* tetrakis(dimethylamino)ethylene.
[a]Reaction conditions: **1a** (0.2 mmol), **2a** (0.3 mmol), **3a** (0.6 mmol), Ni catalyst (10 mol%), ligand (12 mol%), reductant (0.6 mmol), solvent (1 mL), 25 °C, 24 h, N₂ atmosphere.
[b]Ligand (20 mol%) was used.
[c]**1a** (0.3 mmol), **2a** (0.2 mmol) were used.
[d]The reaction run at 35 °C for 36 h.
[e]The isolated yield was shown in parentheses on 0.4 mmol scale.
[f](3-iodopropyl)benzene as the substrate.

30% yield (Table 1, entries 1-6). It should be noted that this reaction is heavily relied on the ligand framework. When other ligands were used, lower reactivity or even no reaction was observed. Then, different catalyst precursors including Ni(II) and Ni(0) were tested using **L4**, Ni(PPh₃)₂Cl₂ represented the best result (Table 1, entries 7-10). We next studied the effect of solvent, such as dimethyl sulfoxide (DMSO), acetonitrile (CH₃CN), ethyl acetate (EtOAc) shut down the reaction completely (Table 1, entries 11-13). Indeed, this reaction can only perform in amide-type solvents, with DMA being the best choice (for details, see the Supplementary Table 3). The observation indicates the solvent plays a critical role in the reaction. Increasing the loading amount of **L4** to 20 mol%, we are glad to find the yield of product **4** can be improved to 60% (Table 1, entry 14). While the ratio of **1a** and **2a** was

changed to 1.5:1, the reaction efficiency was slightly decreased (55%, Table 1, entry 15). Subsequently, other reductants like manganese, tetrakis(dimethylamino)ethylene (TDAE) were screened, less than 5% of product were determined in the crude reaction (Table 1, entries 16 and 17). Finally, the desired yield could be further improved to 79% when the reaction was carried out at 35 °C for 36 h (Table 1, entry 18). The reaction can also proceed well to give desired product **4** in 55% yield choosing (3-iodopropyl)benzene instead of alkyl bromide **1a**. To test the effect of alkene unit for this transformation, a variety of activated olefins were then studied. Unfortunately, all the chosen alkenes, such as acrylate esters, *N*-phenylacrylamide, and (vinylsulfonyl)benzene show much lower reactivity in the standard conditions (<10% yields). The hydro-debromination of alkyl bromide and

**Fig. 2 | The scope of alkyl bromides.** Reaction conditions: **1** (0.4 mmol), **2a** (0.6 mmol), **3a** (1.2 mmol), Ni(PPh₃)₂Cl₂ (10 mol%), **L4** (20 mol%), Zn (1.2 mmol), DMA (2 mL), 35 °C, 36 h, N₂ atmosphere. Isolated yield was shown. [b]**2a** (1.2 mmol)

and **3a** (2.4 mmol) were used, as well as 24% of mono-alkylsilylation obtained. [c]The reaction run at 50 °C for 48 h. [d]Iodocyclopentane as the substrate. d.r. diastereomeric ratio. Boc tert-butyloxycarbonyl, TMS trimethylsilyl, Ts tosyl.

hydroalkylation of alkene are the main byproducts detected in the reaction (for details, see the Supplementary Discussion 2.2 The Investigation of Activated Olefins). These results demonstrated the nitrile group on alkene is crucial for the reaction performance.

## Substrate scope

Under the optimal reaction conditions, the scope and limitation of alkyl bromides was firstly screened, and the result was summarized in Fig. 2. A set of phenyl ring containing primary alkyl bromides, having electron-donating groups (-OMe and -NHBoc) and electron-withdrawing groups (-Cl, -Br and -CN), can be tolerated well in our reductive alkylsilylated reaction, giving the corresponding products 5-9 in good yields (51–76%). It's worth noting that free hydroxyl group on phenyl ring, which is typically not compatible with silylmetal reagent, can also be tolerated in our system (10). The reactions of alkyl bromides linked aromatic heterocycles proceeded smoothly to produce branched alkylsilanes 11-16 in moderate to good yields (39%-74%), including thiophene, indole, benzofuran, carbazole, pyridine and quinoline. Particularly, substrates with coordinating Lewis basic N(sp2) atom (15 and 16) worked well with slightly decreased yields. As expected, primary alkyl bromides bearing linear or cyclic structures are amenable substrates in our system, affording good yields of products (17-20). Remarkably, a plenty of valuable functional units, such as ether (21), halogen atom (22, 23), ester (24), phosphate (25), amide (26), ketone (27), acetal (28), trifluoromethyl (29), nitrile (30), sulfone (31), boronic ester (32), terminal alkene (33), TMS-protected alkyne (34) and free alcohol (35) were all tolerated well in this nickel catalyzed system (20–90% yields), delivering the functionalized organosilicon compounds under mild conditions as well as providing a chance to further modify these molecules to give more versatile structural diversity. When dibromoalkane was selected as the starting materials, the double alkylsilylation reaction occurred as expected to afford bis-

silicon product 36 in 53% yield with 24% of mono-alkylsilylation byproduct 36' observed.

As mentioned, secondary and tertiary alkyl groups failed to be added onto alkene using previously reported carbosilylation method[58], which encourages us to investigate the reactivities of secondary and tertiary alkyl bromides for this reaction. To our delight, reactions of various bromides with either cyclic or acyclic secondary and tertiary carbon as well as iodocyclopentane exhibited good reactivity to give regioisometrically pure products 37-52 in up to 89% yields under the standard conditions. These compounds bearing heterocycles (44-46) and all-carbon quaternary stereogenic center (47-52) offer much more complex and diverse molecular structures compared to the previous protocol involving primary alkyl additions with silylzinc reagent. Given the broad substrate scope and good functionality tolerance of our strategy, we are inspired to explore the developed reaction to modify more complex molecules. Alkyl bromides derived from (−)-perillyl alcohol, (−)-nopol and estrone, successfully underwent the reductive alkylsilylation of acrylonitrile 2a with chlorodimethyl(vinyl)silane 3a to give the silylative derivatives in 37–56% yields (53-55). Anti-inflammatory drugs, such as indomethacin, naproxen and niflumic acid, could be easily transformed to the corresponding alkyl bromides, and exhibited good efficiency in the three-component reaction (56-58). The reactions of antigout drug analogues from probenecid and febuxostat proceeded well under the alkene difunctionalization conditions to deliver silicon-containing skeletons (59, 60) with multiple functional groups, including sulfonamide, ester, nitrile and heterocycle. These outcomes demonstrate the method shown here is interesting and potentially useful.

Next, we explored the reactivity of chlorosilanes by varying substituents on silicon atom in this transformation (Fig. 3). The desired reaction proceeded unsurprisingly to yield the corresponding

**Fig. 3 | The scope of chlorosilanes.** Reaction conditions: 1a (0.4 mmol), 2a (0.6 mmol), 3 (1.2 mmol), Ni(PPh$_3$)$_2$Cl$_2$ (10 mol%), L4 (20 mol%), Zn (1.2 mmol), DMA (2 mL), 35 °C, 36 h, N$_2$ atmosphere. Isolated yield was shown. $^b$The reaction run for 72 h. $^c$The reaction run at 50 °C. d.r. diastereomeric ratio.

organosilicon compound **61** in 77% yield when a long alkyl chain was used instead of methyl group on chlorodimethyl(vinyl)silane **3a**. Chloro(methyl)(phenyl)(vinyl)silane **3c** and its derived chlorosilanes with electron-donating group **3d** and electron-withdrawing group **3e**, were all accommodated in this catalytic environment (**62-64**), and no obvious electronic effect was observed. The reaction of chlorodimethyl(phenyl)silane afforded trialkyl silane **65** in moderate yield. Chlorosilanes with multi-phenyl ring show lower reactivity due to steric hindrance, but the product yields can be improved up to 63% with longer reaction time (**66**, **67**). After that, we moved to study trialkyl chlorosilanes, which are more challenging substrates and frequently indolent as an additive in cross-electrophile coupling reactions. Gratifyingly, the target products can be formed in satisfying yields (**68-71**, 19–82%) using a set of common alkyl chlorosilanes, although the lower yield was obtained in the reaction of triethyl chlorosilane used as the substrate (**71**), which can probably be ascribed to its steric effect.

## Synthetic applications

To further showcase the synthetic utility of the current strategy, a gram-scale reaction of model substrates was carried out under standard conditions, affording desired product **4** in 67% yield (Fig. 4a). The silylated nitrile **4** can undergo desilylation reaction easily in the presence of CsF to give alkylnitrile **72** in good yield (Fig. 4a). The alkene motif in the product is a versatile synthon which can be converted to alkyl bromide **73** and thioether **74** efficiently through addition reactions (Fig. 4a). Surface modification of inorganic solids plays an important effect in material science. We studied the surface modification of glass by incorporation of silylated functional group using our product. As shown in Fig. 4b, after treatment with **4** under iridium catalyst, the glass surface became more hydrophobic, in which the contact angle increased from 31° for the bare surface to 71° for the immobilized surface. In addition, a gram-scale reaction of **1a**, **2a** with chlorotrimethylsilane **3i** can also perform well to provide target product **68** in 69% yield (Fig. 4c). The TMS substituted alkylnitrile **68** can be transformed into β-hydroxy nitrile **75** in 87% yield via desilylative addition with 4-chlorobenzaldehyde (Fig. 4c), which is a useful precursor of β-hydroxycarboxylic acids or β-amino alcohols. Furthermore, **68** could go through [3,3]-sigmatropic rearrangement with diphenyl sulfoxide to synthesize challenging α-aryl primary amide **76** in good yield (Fig. 4c). On the other hand, **68** could also work as a α-cyano alkyl nucleophile and subsequently couple with ethyl 4-bromobenzoate under palladium catalysis to afford α-aryl nitrile **77** in 72% yield (Fig. 4c).

## Mechanistic studies

To gain the preliminary understanding of the mechanism of this reductive alkylsilylative reaction, a set of control experiments were conducted as shown in Fig. 5. The stoichiometric amount of TEMPO as a radical scavenger was added into the model reaction under standard conditions, in which the desired reaction was suppressed (Fig. 5a). (bromomethyl)cyclopropane **1ca** was selected as the reaction partner instead of (3-bromopropyl)benzene **1a** to afford the ring-opening product **78** in 72% isolated yield under optimal conditions without any expected ring-keeping product detected (Fig. 5b). these results suggest alkyl radical is generated from alkyl bromide and involved in this alkene alkylsilylative reaction. Subsequently, stoichiometric reaction of **1a**, **2a** and **3a** in the presence of Ni(COD)₂ (1 equiv.) and **L4** (2 equiv.) without any zinc in DMA at 35 °C for 36 h led to only 3% of target product **4** along with homocoupling and hydro-debromonation of **1a** as byproducts, and alkyl bromide **1a** almost consumed (Fig. 5c). In a separate vessel, the catalytic reaction was run using 10 mol% Ni(COD)₂ and 20 mol% **L4** with 3 equivalents of zinc dust as the reductant to give the alkylsilylative product **3a** in 70% yield (Fig. 5d). these observations indicate: (1) bromide **1a** was almost

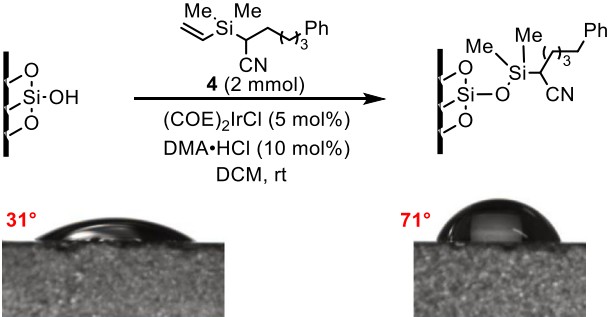

**Fig. 4 | Gram-scale reactions and synthetic applications. a** Gram-scale reaction of **1a**, **2a** and **3a**, and synthetic applications of **4**; (i) **4** (0.2 mmol), CsF (0.048 mmol), CH₃CN (4.3 mL), room temperature, 6 h; (ii) **4** (0.2 mmol), *N*-bromosuccinimide (0.5 mmol), TEMPO (0.4 mmol), CH₃CN (0.4 mL), room temperature, 3 h; (iii) **4** (0.4 mmol), propane-1-thiol (0. 48 mmol), AIBN (0.01 mmol), THF (1 mL), 50 °C, 48 h, N₂ atmosphere; **b** surface modification of glass using **4**, Glass slide (1.5 × 1.5 cm²), **4** (2 mmol), [(COE)₂IrCl]₂ (0.1 mmol), DMA · HCl (0.2 mmol), DCM (1.0 mL), room temperature, 12 h; **c** gram-scale reaction of **1a**, **2a** and **3i**, and synthetic applications of **68**; (iv) 4-chlorobenzaldehyde (0.2 mmol), **4** (0.4 mmol), CsOAc (0.02 mmol), DMF (1 mL), 40 °C, 12 h; (v) sulfinyldibenzene (0.2 mmol), **4** (0.4 mmol), TfOH (0.3 mmol), DCE (2 mL), 50 °C, 18 h; (vi) Ethyl 4-bromobenzoate (0.2 mmol), **4** (0.3 mmol), Pd₂dba₃ (0.01 mmol), Xantphos (0.02 mmol), and ZnF₂ (0.12 mmol) in DMF (0.2 mL), 90 °C, 18 h, N₂ atmosphere. Reaction details were shown in the supplementary notes 3.3 Gram-scale Reactions and Synthetic Applications.

consumed by Ni(0), and the reaction pathway should be involved alkyl bromide reacted with nickel catalyst firstly; (2) the zinc dust is probably engaged in essential reduction step in the catalytic cycle, not just a terminal reductant.

On the basis of our experimental results and previous reports[24, 25, 66], a catalytic cycle is proposed in Fig. 6. We speculate Ni(0)

**(a)** *Radical-trapping experiment*

Ph⌒⌒Br + �̈CN + Cl—Si—Me  → *standard condtions* / **TEMPO (3 equiv)** →  Ph⌒⌒⌒Si(Me)(Me)(vinyl), CN

**1a**      **2a**      **3a**                                            **4, 0%** yield

**(b)** *Radical-clock experiment*

▷—Br + ⌈CN + Cl—Si—Me  → *standard condtions* →  product

**1ca**      **2a**      **3a**                                            **78, 72%** isolated yield

**(c)** *Control experiments*

Ph⌒⌒Br + ⌈CN + Cl—Si—Me  → Ni(COD)₂ (1 equiv) / **L4** (2 equiv) / DMA (0.2 M), 35 °C, 36 h →  product

**1a**      **2a**      **3a**
          (1.5 equiv)  (3 equiv)

**4, 3%** yield

homocoupling of **1a**: 26%
hydro-debromination of **1a**: 1.6%
recovered **1a**: 4%

**(d)**

Ph⌒⌒Br + ⌈CN + Cl—Si—Me  → Ni(COD)₂ (10 mol%) / **L4** (20 mol%) / Zn (3 equiv) / DMA (0.2 M), 35 °C, 36 h →  product

**1a**      **2a**      **3a**
          (1.5 equiv)  (3 equiv)

**4, 70%** yield

**Fig. 5 | Mechanistic studies. a** Radical-trapping experiment; (**b**) radical-clock experiment; (**c**) control experiment; (**d**) control experiment. Reaction details were shown in the supplementary notes 3.4 Mechanistic Studies.

should be generated in-situ via reduction of Ni(II) precursor by zinc. Then, the alkyl bromide **1** reacts with Ni(0) via single electron reduction to give Ni(I) and a alkyl radical; the latter is quickly captured by activated olefin **2a** to produce a new carbon radical **A**. The formed radical **A** recombined with Ni(I) to afford alkyl Ni(II) species **B**, which is subsequently reduced by Zn to yield an alkyl Ni(I) intermediate **C**. The oxidative addition of **C** with chlorosilane **3** generate alkyl silyl Ni(III) complex **D** that undergoes reductive elimination to deliver desired alkylsilylative product with generation of Ni(I), which is reduced to Ni(0) by zinc to enter next catalytic cycle.

In conclusion, a regioselective nickel-catalyzed reductive alkylsilylation of acrylonitrile using more readily available unactivated alkyl bromides and chlorosilanes with good efficiency under mild conditions is discovered. Notably, this approach provides a versatile and powerful tool for synthesizing a wide range of alkylsilanes with more enriched molecular complexity and diversity, with broad substate scope including primary, secondary and tertiary alkyl bromides, as well as Csp2 and Csp3 substituted chlorosilanes. This method shows good functional group tolerance. Moreover, the practicability of this strategy is demonstrated by its application in functionalization of more complex molecules. Given the increased applications of organosilicon compound, we believe our study is an important contribution in

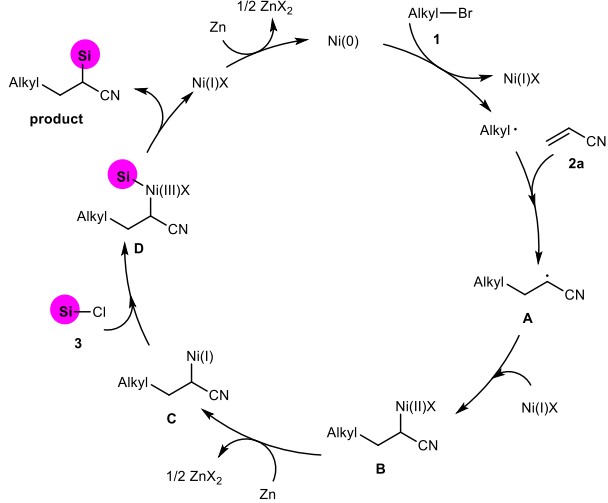

**Fig. 6 | Proposed mechanism.** Proposed catalytic cycle. The silyl groups are colored in pink.

organosilicon chemistry and facilitate current endeavors in designing practical methods for construction of C-Si bonds.

## Methods
### General procedure for Ni-catalyzed alkylsilylation of acrylonitrile
The procedure was conducted in a nitrogen-filled glove box. To a reaction vial equipped with a magnetic stir bar was added Ni(PPh$_3$)$_2$Cl$_2$ (26.4 mg, 0.04 mmol), 2-(pyridin-2-yl)−4,5-dihydrooxazole (12.0 mg, 0.08 mmol), Zn (78.0 mg, 1.2 mmol). A solution of **1** (0.4 mmol), **2a** (40.0 μL, 0.6 mmol) and chlorosilane **3** (1.2 mmol) in DMA (2.0 mL) was then added. The reaction vial was sealed and removed from the glove box. The mixture was stirred at 35 °C for 36 h, subsequently quenched with water (10.0 mL) and extracted with dichloromethane (3 × 15.0 mL). The combined organic layers were washed with water, brine, dried over anhydrous Na2SO4, and concentrated under reduced pressure. The residue was purified by flash chromatography on silica gel to afford product.

## Data availability
The authors declare that the data supporting the findings of this study, including experimental details and compound characterization, are available within the article and its Supplementary Information file. All data are available from the corresponding author upon request.

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

## Acknowledgements

We gratefully acknowledge the financial support of the "Thousand Talents Plan" Youth Program, the "Jiangsu Specially-Appointed Professor Plan" (R2020T30), the "Innovation & Entrepreneurship Talents Plan", the National Natural Science Foundation of China (22101136), the Natural Science Foundation of Jiangsu Province (BK20200806) and the Start-up Foundation for Introducing Talent of Nanjing University of

Information Science and Technology. We also thank Dr. Jiawei Chen (Columbia University) for assistance with the manuscript preparation.

## Author contributions

X.Z. conceived the methodology and wrote the manuscript with the assistance of other authors. J.S., Y.Z., R.G., X.L. and A.L. conducted experiments. All the authors analyzed the data.

## Competing interests

The authors declare no competing interests.
