## [Peer Review File · Nature Communications]

Regioselective Ni-Catalyzed Reductive Alkylsilylation of Acrylonitrile with Unactivated Alkyl Bromides and ChlorosilanesREVIEWER COMMENTS

Reviewer #1 (Remarks to the Author):

In this manuscript, Zhang and co-workers reported a regioselective Ni-catalyzed reductive alkylsilylation of activated olefins with unactivated alkyl bromides and chlorosilanes for construction of alkylsilanes with great structural complexity and diversity. In this work, chlorosilanes were used as the silyl reagents instead of previously employed silylmetal reagents, which can avoid preparing and handling of air and moisture sensitive organometallic reagents and further show much better functional group tolerance. Significantly broad substrate scope was demonstrated with respect to both of alkyl bromides and chlorosilanes. Also, late-stage functionalization of drug relevant molecules, gram scale reaction and application of the products highlight the synthetic potency of this reaction. In addition, a realistic mechanism proposal is given in Figure 7, which seems plausible. Given the importance of organosilicon compounds and the novelty of reductive silylative difunctionalization of alkenes, this efficient and practical protocol is an important contribution in organosilicon chemistry. Therefore, I'm glad to recommend the publication of manuscript in Nat. Commun., after some revisions were addressed.

1. I find that some important works in this field are missing in the citations, in particular from groups that strongly initiated the field of reductive difunctionalization of alkenes such as Nevado and Diao. Selected works should be cited.
2. In the Figure 2, alkyl bromides were tolerated very well, what's about the result of reaction using alkyl iodides?
3. In this manuscript, the authors mentioned cyano group on alkene is crucial for the reaction performance, and only acrylonitrile was shown. What's about the result of reaction choosing other cyano group containing alkenes?
4. In the SI, the "DMA" was used in section 2 directly, which should be provided its full name and relevant information in General information.
5. In the page S12 of SI, compound 1m is colorless oil, but the melting point data is given, which should be corrected. Actually, the melting point data of known compounds are not necessary.
6. In the page S56 of SI, the used volume of DMA for gram-scale reaction of 4 should be doublechecked.
7. In the page S57 of SI, the structure of NBS should be corrected.
8. In the page S60 of SI, the melting point data of compound 76 was missed.

Reviewer #2 (Remarks to the Author):

Zhang reports a method for the carbosilylation of acrylonitrile with alkyl halides and chlorosilanes. The reaction is predicted to work by radical addition to acrylonitrile and capture of the radical with Ni. The resulting Ni-complex is then trapped with the chlorosilane. The reaction works well for a variety of alkyl bromides (primary, secondary, and tertiary) and chlorosilanes.

Prior has demonstrated that carbosilylation is possible. However, the authors are correct in that all known methods use nucleophilic Si. This is problematic as nucleophilic Si is not very easy to work with. pinB-SiMe₂Ph is available, but expensive and difficult to make. Li/Zn-SiMe₂Ph are not functional group tolerant or easy to work with. Given the background, I think the authors have made a notable advance in the synthesis of silicon containing compounds and thus may be appropriate for Nat. Comm.

Before acceptance, I think the authors need to address some issues:

In what situations would someone not just take the alkyl nitrile, deprotonate and trap with the silyl chloride? I think these need to be highlighted and addressed.

The title says activated alkenes, but only acrylonitrile is used. Therefore, the title should be changed to "acrylonitrile."

What happens with other activated alkenes. Getting other things to work would greatly strengthen the study.

Why did the author choose to use diethyl-vinylchlorosilane as opposed to trimethylsilyl chloride? Just curious.

Reaction conditions should be added to Fig. 5.

How do the authors know a Ni-enolate is formed as opposed to a Zn-enolate? Can a Zn-enolate react with the chlorosilane? The authors could test this by using a chiral ligand. If any ee is observed, that suggests that Ni is involved in the reaction with chlorosilane. I think the authors need to do this. Otherwise, the later part of the mechanism seems speculative.

The section on late stage functionalization needs to be removed. Why is the example with Naproxen or Estrone (or most others) considered complex? Did the authors really expect something different to happen with these examples? I am sorry to be critical of this section, but the idea of over selling one's work with "late stage functionalization" with examples that do not demonstrate that is far too common. Late stage functionalization is important, but only when demonstrated on complex systems and the skeleton, C-H bonds, or functional groups of the original molecule are modified. Is it not significant at all to make a derivative of a non-complex molecule and claim late stage functionalization. These examples should either be removed and placed with the rest of the substrate scope. For example, this statement, "These outcomes give solid evidence of the practicability of our method, and should be potentially applicable in drug discovery" is not true. The method shown here is interesting and potentially useful, but not for late stage functionalization.

"forbidden completely" change to "suppressed."

Point-by-point responses to the comments

Reviewer: 1

Recommendation: recommend the publication of manuscript in Nat. Commun., after some revisions were addressed.

Comments:

1. I find that some important works in this field are missing in the citations, in particular from groups that strongly initiated the field of reductive difunctionalization of alkenes such as Nevado and Diao. Selected works should be cited.

Response: We have cited selected works as references 73 and 74 in the revised manuscript.

2. In the Figure 2, alkyl bromides were tolerated very well, what's about the result of reaction using alkyl iodides?

Response: Thanks a lot for the comment. We selected (3-iodopropyl)benzene and iodocyclopentane as the substrates for the three-component reaction under standard conditions, which gives the desired products in 55% and 54% yields, respectively. The results have been added in the revised manuscript.

3. In this manuscript, the authors mentioned cyano group on alkene is crucial for the reaction performance, and only acrylonitrile was shown. What's about the result of reaction choosing other cyano group containing alkenes?

Response: We tried (*E*)-but-2-enenitrile and cinnamionitrile instead of acrylonitrile under standard conditions. Both of them are not compatible in this

system and no target products were observed.

4. In the SI, the “DMA” was used in section 2 directly, which should be provided its full name and relevant information in General information.

Response: We have added the full name of “DMA” as “N,N-dimethylacetamide” in the section of General information.

5. In the page S12 of SI, compound 1m is colorless oil, but the melting point data is given, which should be corrected. Actually, the melting point data of known compounds are not necessary.

Response: Thanks for the suggestion. The melting point data was removed from revised SI.

6. In the page S56 of SI, the used volume of DMA for gram-scale reaction of 4 should be doublechecked.

Response: Thanks for pointing this out. We have made the correction in the revised SI.

7. In the page S57 of SI, the structure of NBS should be corrected.

Response: We have made the corresponding correction in the revised SI.

8. In the page S60 of SI, the melting point data of compound 76 was missed.

Response: We have added the data in the revised SI.

Reviewer: 2

Recommendation: appropriate for Nat. Comm. Before acceptance, I think the

authors needs to address some issues.

Comments:

1. In what situations would someone not just take the alkyl nitrile, deprotonate and trap with the silyl chloride? I think these need to be highlighted and addressed.

Response: According to the literatures, the product reported in this work can also be synthesized through the reaction of alkyl nitrile deprotonated by organolithium reagent and subsequently trapped by silyl chloride. However, the use of organometallic reagent limited the substrate scope and functionality tolerance largely. In contrast, our method is more general, which can accommodate a much broader range of functional groups, such as ketone, sulfone and hydroxyl group *et al.* The relevant statement “Although this type of product can also be achieved through nucleophilic substituted reaction of deprotonated alkyl nitrile with silyl chloride, organolithium compound used as the base leads the reaction exhibiting poor functional group tolerance and restricted application.” has been added in the revised manuscript.

2. The title says activated alkenes, but only acrylonitrile is used. Therefore, the title should be changed to “acrylonitrile.”

Response: We have changed the title to “Regioselective Ni-Catalyzed Reductive Alkylsilylation of Acrylonitrile with Unactivated Alkyl Bromides and Chlorosilanes”.

3. What happens with other activated alkenes. Getting other things to work would greatly strengthen the study.

Response: Thanks for the constructive suggestion. Actually, we have shown some other activated alkenes such as acrylate esters, *N*-phenylacrylamide, and (vinylsulfonyl)benzene in current manuscript (after the section of condition optimization), and details can be found in the SI. All of them exhibited much lower reactivity under standard conditions (<10% yields). Moreover, based on the reviewer 1’s comments, we also tried (*E*)-but-2-enitrile and cinnamitrile under optimal conditions, and no desired products were observed. The results have been added in the revised SI. The observation from these experiments

indicated the structure of alkene is critical for the reaction performance.

4. Why did the author choose to use dimethyl-vinylchlorosilane as opposed to trimethylsilyl chloride? Just curious.

Response: We do show a number of trialkylsilyl chlorides including trimethyl silyl chloride in the manuscript and the results have been summarized in Figure 3. The reason choosing dimethyl-vinylchlorosilane as the model substrate is mainly due to its versatility as a synthetic synthon that could be further converted to different functional groups in a latter stage.

5. Reaction conditions should be added to Fig. 5.

Response: We have added the reaction conditions as footnotes to the Figure.

6. How do the authors know a Ni-enolate is formed as opposed to a Zn-enolate? Can a Zn-enolate react with the chlorosilane? The authors could test this by using a chiral ligand. If any ee is observed, that suggests that Ni is involved in the reaction with chlorosilane. I think the authors need to do this. Otherwise, the later part of the mechanism seems speculative.

Response: Thanks a lot for the great suggestion. We screened a variety of chiral ligands including different oxazolines and diamines, and found that ligand (1R,2R)-N1,N2-Bis(2-(bis(3,5-dimethylphenyl)phosphino)benzyl)cyclohexane-1,2-diamine gave the product with 12% ee. Combined the result with previous reports (*Angew. Chem. Int. Ed.* 2021, 60, 18587–18590; *Org. Lett.* 2021, 23,

7855–7859), we believe the Ni-enolate cycle is a favorable pathway for the mechanism. In addition, encouraged by the promising result, the investigation on the asymmetric version of this reaction is ongoing and will be discussed in detail in a separate work in the future.

HPLC analysis was performed on Shimadzu SPD-20A using Daicel Chiralpak OD Column. Chiral HPLC Analysis Conditions: a) Column: CHIRALCEL®OD-H, 4.6mm Φ *250mmL, 5 μm ; b) Mobile phase: n-Hexane/ IPA = 90/10 (V/V); c) Flow rate: 1.0 mL/min; d) Abs. detector: 220 nm

1 Det.A Ch1/220nm

PeakTable

Peak#	Ret. Time	Area	Height	Area %	Height %
1	6.663	8532968	667904	49.650	42.144
2	7.781	8653367	916923	50.350	57.856
Total		17186335	1584827	100.000	100.000

Figure 1. Chiral HPLC chromatogram for racemic **79**

Figure 2. Chiral HPLC chromatogram for enantioenriched 79

7. The section on late stage functionalization needs to be removed. Why is the example with Naproxen or Estrone (or most others) considered complex? Did the authors really expect something different to happen with these examples? I am sorry to be critical of this section, but the idea of over selling one's work with "late stage functionalization" with examples that do not demonstrate that is far too common. Late stage functionalization is important, but only when demonstrated on complex systems and the skeleton, C-H bonds, or functional groups of the original molecule are modified. Is it not significant at all to make a derivative of a non-complex molecule and claim late stage functionalization. These examples should either be removed and placed with the rest of the substrate scope. For example, this statement, "These outcomes give solid evidence of the practicability of our method, and should be potentially applicable in drug discovery" is not true. The method shown here is interesting and potentially useful, but not for late stage functionalization.

Response: We removed the section of late-stage functionalization and the examples were added in the substrate scope of Figure 2. And the statement "These

outcomes give solid evidence of the practicability of our method, and should be potentially applicable in drug discovery” has also been changed to “These outcomes demonstrate the method shown here is interesting and potentially useful.”

8. “forbidden completely” change to “suppressed.”

Response: Thanks for the suggestion. We have made the change in the revised manuscript.

REVIEWERS' COMMENTS

Reviewer #1 (Remarks to the Author):

The authors have already addressed all the questions I raised and revised the manuscript and supplementary materials accordingly. Therefore, I support it to publish without further revision.

Reviewer #2 (Remarks to the Author):

The authors most addressed my concerns.

However, the section on "Natural products and pharmaceutical" is still present.

I am sorry, but it very frustrating the see members of the chemical community try to oversell their work. To many chemists try to sell their work in this way, and I think it hurts our field.

Even the statement on lines 195-196 is overselling. "Great functional group tolerance". Really? The function groups shown are nice, but there is much room for improvement

Point-by-point responses to the comments

Reviewer: 1

Comments:

The authors have already addressed all the questions I raised and revised the manuscript and supplementary materials accordingly. Therefore, I support it to publish without further revision.

Response: Thanks again for the reviewer's useful suggestions and affirmation on our work.

Reviewer: 2

Comments:

The authors most addressed my concerns.

However, the section on "Natural products and pharmaceutical" is still present.

I am sorry, but it very frustrating the see members of the chemical community try to oversell their work in this way, and I think it hurts our field.

Even the statement on lines 195-196 is overselling. "Great functional group tolerance". Really? The function groups shown are nice, but there is much room for improvement

Response: Thanks for the reviewer's comments, the section "Natural products and pharmaceutical" has been changed to "other alkyl bromides", and the statement on lines 195-196 as well as other places involved "great functional group tolerance" has also been modified to "good functional group tolerance".